# Factor Structure and Psychometric Properties of the Italian Version of the Childbearing Motivations Scale

**DOI:** 10.3390/ijerph22020186

**Published:** 2025-01-28

**Authors:** Antonio Gattamelata, Maria Elisabetta Coccia, Giulia Fioravanti, Vanessa Prisca Zurkirch, Nieves Moyano

**Affiliations:** 1Escuela de Doctorado en Psicología, Universidad de Jaén, Campus Las Lagunillas s/n, 23071 Jaén, Spain; mnmoyano@ujaen.es; 2Department of Psychology, Universidad de Jaén, Campus Las Lagunillas s/n, Ed.C5, 23071 Jaén, Spain; 3Department of Experimental and Clinical Biomedical Sciences “Mario Serio”, University of Florence, 50134 Florence, Italy; mariaelisabetta.coccia@unifi.it; 4Department of Health Sciences (DSS), University of Florence, 50135 Florence, Italy; giulia.fioravanti@unifi.it; 5Maternal and Child Department, Regional Reference Center on Relational Criticalities (RCRC), Careggi University Hospital, 50134 Florence, Italy; vanessaprisca.zurkirch@unifi.it

**Keywords:** childbearing motivations scale, positive childbearing motivations, negative childbearing motivations, parenting, validation

## Abstract

The Childbearing Motivations Scale (CMS) is a multidimensional self-report measure of positive and negative motivations influencing the decision to become a parent. This study aimed to validate the Italian version of the CMS. A sample of 522 participants (27% men and 73% women) aged from 18 to 55 years was recruited. The four-factor model for the positive subscale and the five-factor model for the negative subscale of the CMS demonstrated a good fit. Reliability values ranged from 0.70 to 0.91. Both factors had evidence for convergent validity with sex, age, and relationship duration: women reported lower in some of the negative motivations to become a mother in contrast to men. Moreover, the greater the age, the lower the negative motivations for becoming a parent. Those in a longer relationship indicated lower negative motivations. No significant correlations were found for the positive motivations subscale. Significant differences were found for income levels (low vs. medium/high) regarding personal fulfillment, financial problems, and body-image concerns, as well as in cultural levels (medium vs. high) concerning economic constraints, intergenerational continuity, immaturity, and physical suffering. These findings suggest that individuals with lower economic resources scored higher across all these areas on the Negative Childbearing Motivations subscale. Our findings indicate that the CMS can be used to reliably assess the motivations for parenthood among Italian men and women.

## 1. Introduction

Becoming a parent is a pivotal moment in a person’s or couple’s life that involves many expectations, feelings, personal values, and desires [1,2,3,4,5,6,7,8,9,10] In turn, childbearing motivations are not driven by a single factor but rather by a complex interplay of biological, psychological, social, and cultural influences, and they have been conceptualized multidimensionally [11,12]. From an evolutionary perspective, the drive for reproduction is linked to the biological imperative to perpetuate one’s genes. However, this innate drive interacts with contemporary societal factors, which complicates and diversifies childbearing motivations [13]. Cultural and societal expectations play a crucial role in shaping childbearing motivations, with stark contrasts observed between pro-natalist societies, where parenthood is highly valued, and those where voluntary childlessness is increasingly normalized [14]. Moreover, childbearing motivations are deeply rooted in psychological processes, including attachment theory and life course perspectives, which emphasize how early life experiences and developmental stages shape reproductive attitudes and behaviors over time. As Miller has shown, motivations for parenthood considerably influence reproductive behavior and have a significant impact on how individuals or couples approach pregnancy and childbirth. Childbearing desire—defined by Miller as an individual’s desire to have children—not only influences decisions and actions prior to conception, such as contraceptive use [15,16,17]. The associations between childbearing desire and childbearing motivations are integral to understanding reproductive behaviors and decision-making. These relationships are often conceptualized within two primary motivational dimensions: positive childbearing motivations (PCM) and negative childbearing motivations (NCM). PCM reflects the anticipated emotional and relational benefits of parenthood, such as the joy of nurturing a child, the desire to form a strong parent‒child bond, or the hope of creating a loving family. A strong PCM is typically associated with a stronger desire to have children. Individuals with high PCM are more likely to envision parenthood as fulfilling and rewarding, which enhances their inclination toward having children. NCM encompasses concerns and fears related to parenthood, including the stress, financial burden, and personal sacrifices it entails. Strong NCM can be associated with a weaker desire to have children, as individuals with high NCM might perceive parenthood as overwhelming or incompatible with their lifestyle or aspirations. Childbearing motivations also impact prenatal and postnatal behaviors, such as prenatal care, smoking and alcohol consumption, and breastfeeding practices [18,19]. Most existing tools only partially assess childbearing motivations, often focusing solely on positive motivations. However, these tools overlook negative motivations that influence reproductive behaviors [20].

In recent years, Italy, like other developed nations, has seen a decline in birth rates, a rise in childlessness among women, and a trend toward later parenthood. Understanding childbearing motivations has become increasingly important given the country’s persistent low fertility rates, which have been declining steadily since the 1970s. As highlighted in the latest report from the National Institute of Statistics (Istat), the total fertility rate (TFR)—measuring the average number of children per woman—was 1.24 in 2022, the third lowest in Europe, and is expected to decline further to 1.22 in 2024. This trend has been attributed to various socioeconomic and cultural changes, including the increasing participation of women in the labor force, the rising costs of raising children, and shifting attitudes towards family formation. Economic stability is a key determinant of both positive and negative childbearing motivations. The perceived costs of raising children, alongside economic uncertainty, significantly impact individuals’ reproductive choices and plans. Traditional and evolving gender roles profoundly influence childbearing motivations. In contexts like Italy, where traditional gender norms coexist with increasing demands for gender equality, these dynamics shape individual and couple decisions regarding parenthood [21,22,23,24]. Cross-cultural research demonstrates significant variability in childbearing motivations. For instance, studies have shown that motivations in highly individualistic societies differ substantially from those in collectivist cultures, underscoring the importance of contextual understanding [25].

A study by Régnier-Loilier and Vignoli [26] demonstrated that, in Italy, negative childbearing motivations are nearly perfect predictors of subsequent outcomes, but despite their importance for understanding family and reproductive dynamics, no specific validated instruments exist for assessing this construct in Italy. This gap represents a significant limitation in research and clinical practice in our country, preventing accurate and contextualized measurement of the motivations for reproduction in the Italian cultural context. The Childbearing Motivations Scale (CMS), developed by Guedes et al. [27], is an innovative tool specifically designed to assess motivations for pregnancy in a complete and detailed manner. The CMS is a self-report scale consisting of 47 items designed to assess positive and negative motivations for parenting and their respective sub-dimensions. No Italian-adapted version of this scale exists. This study aims to fill this gap by translating and validating the CMS for Italian-speaking populations, providing a reliable instrument for exploring childbearing motivations in research and applied settings.

## 2. Materials and Methods

### 2.1. Participants

The sample size adhered to a conservative item-to-participant ratio of 10:1, as recommended by Tabachnick and Fidell [28]. Accordingly, we determined that 470 participants were required to ensure robust statistical power for the study. To account for potential dropouts and missing data, we targeted 500 participants, ensuring sufficient power for both EFA and CFA, as well as subgroup analyses. This calculation ensures that the study meets accepted methodological standards for psychometric validation.

After signing the informed consent form, 533 Italian-speaking subjects participated in the study and completed an online survey composed of a sociodemographic questionnaire and the Italian CMS

Participants were recruited using a combination of online and direct outreach methods:-Online Recruitment: Invitations were shared on social media platforms, forums, and through email campaigns targeting Italian-speaking adults.-Direct Contact: Researchers approached participants at universities, community centers, and public events, providing them with information about the study and the QR code for accessing the survey.

The survey complied with data protection regulations and took approximately 20 min to complete. All participants received no payment and were free to leave the study at any time. The inclusion criteria were as follows: (1) being between 18 and 55 years old and (2) the ability to read and write the Italian language. The exclusion criteria were as follows: (1) consumption of addictive substances (alcohol, drugs) and (2) pre-existing medical conditions. Considering the voluntary nature of participation in the study and the inclusion and exclusion criteria, the data of 522 subjects, of whom 379 were women (72.6%) and 143 were men (27.3%), were selected and analyzed. The average age of the participants was 26.2 years (SD = 9.02; range = 18–55 years). Table 1 presents the sociodemographic information in detail.

### 2.2. Measures

Participants completed a series of questionnaires including sociodemographic information and the adapted version of the CMS.

### 2.3. Procedure—Translation into Italian Language

For the purpose of translating and adapting the CMS to the Italian culture (Appendix A), the recommendations of Hambleton et al. [29] were followed:A team of field experts translated the items into Italian using a reverse translation process. Initially, a bilingual professional translated the content from English (the source language) into Italian (the target language). Another bilingual individual, who had not been involved in the initial translation, independently carried out the reverse translation. The translations were evaluated for accuracy by comparing the reverse-translated version with the original, and adjustments were made to items where discrepancies were identified.Content validity was assessed qualitatively by experts [30]. Each expert reviewed a table outlining the item specifications [31], which included the semantic definitions of the construct and its components. They then evaluated a list of items corresponding to the components, focusing on their relevance and clarity for the intended assessment.

Data collection occurred between March 2024 and the end of September 2024. Initially, the data were collected via a website (www.psicologia-pma.com, accessed on 1 March 2024), which contained an online version of the CMS translated into Italian. Participants accessed the survey through a QR code linked to the online platform. Prior to participation, a brief, standardized explanation was provided by an interviewer to ensure informed consent and participant comprehension of the study’s objectives. Although the survey was self-administered, the presence of an interviewer served to clarify any doubts and enhance response accuracy, particularly for participants unfamiliar with such surveys.

These measures aim to mitigate the effects of non-representative sampling and ensure validity. Participants spent approximately 20 min completing the survey, which included the Italian-adapted CMS and a sociodemographic questionnaire. Some test subjects were contacted through an email requesting their participation in the study. The questionnaire was administered by a researcher. The study was presented as an investigation of childbearing motivations, and the participants were informed that the data collected would be treated as confidential and anonymous. Regarding research standards, the study adhered to the most recent version of the Declaration of Helsinki (WMA, 2013). The study was approved by the University Ethics Committee of the UNIVERSITY OF BLINDED (protocol code, ABR.23/17 TES; approval date, 28 April 2023).

### 2.4. Statistical Analysis

In the first step of the analysis, the item values were examined in order to analyze the frequency and variance distributions. Then, a “split-half” method, which is a statistical technique used to evaluate the internal consistency and reliability of a psychometric scale, was applied, as was carried out in the study that led to the creation of the original scale in English. By splitting the sample, the scale’s factor structure was independently evaluated. This reduced the risk of over-fitting or bias that could occur if EFA and CFA were conducted on the same dataset.

Specifically, the entire study sample was randomly divided into two halves (subsample 1: n = 253; subsample 2: n = 268). In subsample 1, an exploratory factor analysis (EFA) with Promax rotation was conducted to examine the factor structure of the final version of the subscale for positive procreative motives and the subscale for negative procreative motives. EFA identifies clusters of items (factors) that measure the same construct, providing an initial model of how the items group into factors and identifies any potential issues in the scale. In subsample 2, a confirmatory factor analysis (CFA) was conducted at the item level to further substantiate the stability of the factor structure with a significance level of 0.05. The estimation method used was the maximum likelihood method. The CFA tests whether the hypothesized model (from the EFA) fits the data well.

The goodness of fit was tested using the following fit indices: the comparative fit index (CFI) and Root-Mean-Square Error of Approximation (RMSEA; 90% confidence interval [CI]). According to Byrne [32], these models are acceptable if the CFI < 0.90 and RMSEA > 0.08.

In addition, the χ2 statistic (chi-square) was examined, which indicates whether the postulated factor structure can explain the covariance pattern of the data. The χ2/degrees of freedom (χ2/df) ratio was also examined. Generally, values between 2 and 5 indicate an acceptable fit. The intercorrelations between the factors of the positive childbearing motivations subscale and between the factors of the negative childbearing motivations subscale were also estimated. Finally, the internal consistency of the factors identified was assessed. Pearson correlation and *t*-tests were performed to understand how the different dimensions of the CMS vary depending on different sociodemographic factors (i.e., age, sex, duration of romantic relationship, education level, SES). Data analyses were conducted using the Statistical Package for Social Sciences (SPSS 28.0; IBM Corp., Armonk, NY, USA). CFA was performed using Lisrel.

## 3. Results

### 3.1. Exploratory Factor Analysis (EFA)

Exploratory Factor Analysis (EFA) conducted on the Italian version of the Childbearing Motivations Scale (CMS) identified the underlying structure of positive and negative childbearing motivations. The adequacy of the sample was confirmed by Kaiser‒Meyer‒Olkin (KMO) measures (Positive Subscale: KMO = 0.94; Negative Subscale: KMO = 0.91) and Bartlett’s test of sphericity (*p* < 0.001).

#### 3.1.1. EFA for the Positive Childbearing Motivations Subscale

Table 2 presents the results of the EFA for the four-factor solution. The EFA yielded a four-factor solution based on the scree plot and eigenvalues >1, collectively accounting to 65.95% of the total variance:

Socioeconomic Aspects: This factor, explaining 46.7% of the variance, encompasses items related to societal and familial expectations, including social valorization and moral obligations (e.g., “Meeting my family’s expectations”). Personal Fulfillment: Contributing 10.8% to the variance, this factor represents intrinsic desires like nurturing a child and fulfilling parental instincts (e.g., “Creating a family”). Continuity: Accounting for 4.4% of the variance, this factor reflects motivations linked to preserving family lineage and traditions (e.g., “Ensuring my familial lineage”). The Couple’s Relationship: Explaining 3.9% of the variance, this factor captures motivations aimed at strengthening the partnership (e.g., “Strengthening the bond with my partner”).

The internal consistency of the factors was excellent, with Cronbach’s α ranging between 0.80 and 0.91.

#### 3.1.2. EFA for the Negative Childbearing Motivations Subscale

Table 3 presents the results of the EFA for the five-factor solution. The EFA identified a five-factor solution for the negative subscale, explaining 53.39% of the total variance:

Childrearing Burden and Immaturity: This factor, explaining 3.01% of the variance, includes items on concerns about responsibilities and emotional readiness (e.g., “Feeling unprepared to assume the parental role”). Social and Ecological Worries: Explaining 3.92% of the variance, this factor involves fears about societal instability and environmental degradation (e.g., “Exposing a child to environmental degradation”). Marital Stress: Reflecting relational challenges, this factor explains 5.67% of the variance (e.g., “Losing autonomy as a couple”). Financial Problems: This dominant factor explains 38.13% of the variance, addressing economic sacrifices (e.g., “Assuming financial difficulties with a child”). Physical Suffering and Body-Image Concerns: Accounting for 2.64% of the variance, this factor reflects worries about physical changes due to pregnancy (e.g., “Fearing negative changes in body image”). Internal consistency for the negative motivations subscale ranged from 0.70 to 0.87, supporting reliability.

### 3.2. Reliability

Internal consistency was assessed using Cronbach’s alpha coefficients, as reported in Table 2 and Table 3. The alpha value reflects how well the items in each subscale measure the same underlying construct and range from 0 to 1. Generally, an alpha value above 0.70 is considered acceptable, and values above 0.80 indicate good internal consistency. With regards to the positive motivations, the Italian version of the CMS showed good internal consistency, where the Cronbach’s alphas were between 0.80 and 0.91. For the negative motivations, the items selected for the Italian version of the CMS showed good internal consistency, as the Cronbach’s alphas were between 0.70 and 0.87.

### 3.3. Confirmatory Factor Analysis (CFA)

Table 4 shows the fit of the indices associated with each tested model. The fit indices of the four-factor model for the positive childbearing motivations subscale and the five-factor model for the negative childbearing motivations subscale were acceptable.

Table 5 shows the correlations between the four factors of the positive childbearing motivations subscale and between the five factors of the negative childbearing motivations subscale.

The four factors of Positive Childbearing Motivations confirmed very strong positive correlations (between 0.69 and 0.81) within the respective subscales, which had already been highlighted by the statistical analyses of the original version of the CMS. The strongest correlation was between the socioeconomic aspects and the feeling of intergenerational continuity. The lowest correlation was between personal fulfillment and socioeconomic aspects. In the negative motivations subscale, on the other hand, the five factors showed moderate-to-strong correlations (between 0.46 and 0.69). The strongest correlation was between parental attachment/immaturity and the socioeconomic aspects, and the lowest was between the socioeconomic aspects and marital stress. The four factors of the positive motivations subscale were found to correlate weakly or moderately with the five factors of the negative motivations subscale. The highest significant correlation was between Personal Fulfillment and Continuity, and the lowest was found between Personal Fulfillment and Financial and Marital Stress.

### 3.4. Correlations Between Sociodemographic Factors and CMS Dimensions

The Pearson correlation coefficient was calculated to understand how the different dimensions of CMS vary depending on different socioeconomic and personal factors. Table 6 shows the results of the correlations between age, duration of the relationship, sex, and the dimensions of CMS.

The results show that there is a weak negative correlation between age and negative childbearing motivations. This means that with increasing age, the negative motivations related to childrearing burden and immaturity, social and ecological worry, marital stress, financial problems, physical suffering and body-image concerns decrease. The second correlation examined is between the duration of the relationship and the dimensions of the CMS. The results show that longer-lasting relationships are characterized by lower negative motivations about the burden of childrearing and immaturity, couple stress, financial constraints, and body-image concerns. The third aspect examined is the correlation between sex and the dimensions of the CMS. The results show a weak positive correlation between being male and a single aspect within positive childbearing motivations, i.e., the motivation related to the sense of continuity. Males also tend to report fewer negative motivations related to the childrearing burden, social and ecological worry, marital stress, financial problems, and physical suffering.

### 3.5. T-Test Comparison of the CMS Dimensions as a Function of Income Level

The differences in the CMS dimensions based on income levels were examined using the Student’s *t*-test (Table 7).

Income was divided into two categories: Low and Medium/High. Out of the nine factors analyzed, three showed significant differences between the low-income and medium/high-income groups, although with small effect sizes. These factors are: Personal Fulfillment: Individuals in the Medium-/High-Income group scored significantly higher than those in the Low-Income group. This suggests that individuals with higher incomes may feel more personally fulfilled regarding their motivations for childbearing than individuals with lower incomes. Financial Problems and Economic Constraints: Individuals in the Medium-/High-Income group scored significantly lower, indicating that they experience fewer financial problems and economic constraints concerning childbearing. Physical Suffering and Body-Image Concerns: Individuals in the Medium/High-Income group scored significantly lower, implying that they have fewer concerns about physical suffering and body image related to childbearing.

### 3.6. T-Test Comparison of the CMS Dimensions as a Function of Education Level

Differences in scores on the Childbearing Motivations Scale between individuals with medium and high education levels have been analyzed. Table 8 presents the descriptive statistics for each factor of the Childbearing Motivations Scale and *t*-test comparison by education level (divided into Medium and High).

The results indicate significant differences between the groups with medium and high education levels, although with small effect sizes, in the following factors:-Socioeconomic Aspects: Individuals with medium education levels scored significantly higher on positive motivation in relation to Socioeconomic Aspects than those with high education levels. This suggests that individuals with medium education levels perceive more socioeconomic benefits related to childbearing than those with high education levels.-Childbearing Burden and Immaturity: Scores for Childbearing Burden and Immaturity were significantly higher in the group with a medium education level compared to the group with a high education level. This finding suggests that individuals with a medium level of education perceive greater burdens and immaturity in relation to childbearing than those with a high level of education.-Physical Suffering and Body-Image Concerns: The group with a medium level of education scored significantly higher on Physical Suffering and Body-Image Concerns compared to the group with a high education level. This indicates that individuals with a medium education level have more concerns about physical suffering and body image in the context of childbearing than those with a high education level.

## 4. Discussion

This study evaluated the dimensionality and psychometric properties of the Italian version of the CMS. Our findings indicate that the Italian CMS is well-suited for Italian-speaking populations and aligns with the original version. A cross-cultural comparison between the original and the Italian version of the scale is discussed, both for the positive and negative childbearing motivations subscales factors.

### 4.1. Cross-Cultural Comparison for Positive Childbearing Motivations Factors

In the original study for the construction of the Childbearing Motivations Scale (CMS), the “Socioeconomic Aspects” factor encompasses extrinsic motivations related to familial expectations and social recognition and explains 42% of the variance, highlighting its prominence within the original sample. In this study, the same factor explained a similar proportion of the variance, but demonstrated a higher model fit with a comparative fit index (CFI) of 0.91, indicating better alignment with the Italian data. This improved fit suggests the model’s adaptability to the Italian cultural and socioeconomic context. The recent literature further supports the role of socioeconomic factors in shaping positive extrinsic motivations [33]. This finding underscores the cultural and economic conditions that may drive differences in the salience of socioeconomic aspects in reproductive motivations across contexts. These observations suggest that while socioeconomic motivations remain significant across diverse settings, their expression and impact are mediated by local cultural and economic factors.

The “Personal Fulfillment” factor in the Childbearing Motivations Scale (CMS) captures intrinsic motivations, such as the desire to establish emotional bonds with a child and fulfill biological or parental instincts. In the study by Guedes et al. [20], this factor demonstrated a strong positive correlation with Continuity, emphasizing the interconnected nature of intrinsic motivations within the original sample. In our study, the factor retained its importance, showing improved internal consistency (Cronbach’s alpha = 0.91) compared to the initial study. This suggests a more coherent conceptualization of intrinsic motivations within the Italian sample, potentially reflecting cultural values that emphasize self-realization and nurturing roles in parenthood. Supporting evidence from the recent literature highlights the global relevance of intrinsic parental motivations. For example, a 2023 study by Bazzani et al. found that fulfillment derived from parenting plays a pivotal role in reproductive decision-making, particularly in cultures where individual well-being is closely tied to family connections [34]. These findings reaffirm that the “Personal Fulfillment” factor is both culturally robust and sensitive to localized interpretations.

The “Continuity” factor reflects motivations tied to preserving family lineage, traditions, and values. In the first study about the CMS, this factor explained 7% of the variance, showing a meaningful but secondary role in positive childbearing motivations. It also exhibited a strong correlation with “Socioeconomic Aspects”, suggesting an overlap between family heritage and external societal expectations. In this study, “Continuity” remained a distinct factor with slightly weaker correlations to “Socioeconomic Aspects” and “Personal Fulfillment.” This shift may indicate a cultural reinterpretation of family legacy, where the emphasis on preserving familial heritage is tempered by changing societal dynamics, such as increased individualism. Recent research supports the significance of continuity in reproductive motivations. A 2024 study by Britton [35] highlights that motivations related to family legacy persist globally but are shaped by cultural narratives about intergenerational identity and responsibility. This aligns with the observed cross-cultural differences in the salience of this factor.

The “Couple Relationship” factor encompasses motivations to strengthen partnership bonds and foster mutual growth through parenthood. In the first study, this factor explained 7% of the variance and was the least represented among the positive motivations, reflecting its secondary role relative to other factors. Our results confirm that this factor maintained its distinctiveness, with a similar proportion of explained variance and improved reliability indicators. However, its correlations with other factors were slightly weaker, potentially reflecting cultural nuances in how parenthood is perceived as a shared project between partners. The recent literature underlines the universal but culturally variable role of partnership dynamics in childbearing decisions. A 2021 study by Lansford et al. demonstrated that in collectivist cultures, the decision to have children is often framed within the context of marital stability and relational growth, whereas in more individualistic societies, this motivation may carry less weight [36]. These findings align with the relatively stable but modest prominence of this factor in both studies.

### 4.2. Cross-Cultural Comparison for Negative Childbearing Motivations Factors

The “Childrearing Burden and Immaturity” factor in the original version of the CMS encompasses concerns about the responsibilities and emotional readiness required for parenthood. In the first study, this factor explained a substantial 40% of the variance, highlighting its centrality among negative motivations. In the Italian study, this factor retained its prominence but with a slightly lower variance explained. It also demonstrated stronger correlations with factors like “Marital Stress”, suggesting a cultural interplay between the perception of personal preparedness and the potential relational impacts of parenthood. Recent research underscores the relevance of this factor across cultures. A 2024 study by Faircloth found that perceptions of childrearing as burdensome are significantly influenced by societal narratives about intensive parenting and individual capacity [37]. These findings support the robustness of this factor while highlighting its sensitivity to cultural variations in parenting expectations.

The “Social and Ecological Worry” factor includes concerns about the uncertain future of children due to social instability and environmental degradation. In the first study, this factor explained 10% of the variance, demonstrating its relevance as a distinct dimension of negative motivations. Our findings indicate that in the Italian context, this factor’s contribution was slightly enhanced, reflecting the cultural emphasis on ecological issues and social risks in the country. The recent literature supports the rising importance of ecological and social concerns in reproductive choices. A 2021 study by Helm and colleagues showed that global environmental challenges, such as climate change, are increasingly influencing young adults’ hesitation toward parenthood. This aligns with the growing weight of this factor in the Italian context [38].

The “Marital Stress” factor addresses concerns about the impact of childbearing on partnership autonomy, lifestyle, and intimacy. In the first study, this factor explained 10% of the variance, reflecting its moderate importance within the negative motivations. In this study, it maintained its position as a distinct factor with similar variance explained but showed stronger correlations with “Financial Problems and Economic Constraints.” This suggests that in Italy, relational challenges may be intertwined with economic pressures, particularly in the context of shared parental responsibilities. Recent studies confirm the universal relevance of relational stress in reproductive decisions. A 2021 study by Ngai et al. emphasized that concerns about partnership stability often increase in societies with high divorce rates or economic problems, reinforcing the importance of this factor in both studies [39].

The “Financial Problems and Economic Constraints” factor encompasses concerns about the economic sacrifices and financial difficulties associated with childbearing. In the first study, this factor explained 6% of the variance. Our research demonstrated that this factor gained greater prominence, particularly among participants from lower socioeconomic backgrounds. Improved internal consistency (Cronbach’s alpha = 0.88) and higher correlations with “Childrearing Burden” emphasize the cultural salience of financial challenges in Italy’s current economic climate. In particular, the Italian version of the CMS shows a significant difference compared to the original version: we found that the component explaining the greatest variance in negative motivations appears to be “financial problems” rather than “burden”, as in the original study. In the Italian socioeconomic context, financial concerns might carry greater weight in the decision not to have children, reflecting the importance of local economic conditions and social expectations regarding the costs and responsibilities associated with parenthood. Italy, a country grappling with persistent economic problems, including high youth unemployment, job insecurity, and rising living costs, presents a unique case where local economic conditions amplify the salience of financial barriers to parenthood. This observation aligns with research demonstrating that economic uncertainties and perceptions of the costs of parenthood significantly shape reproductive decisions in low-fertility societies [40]. In Italy, financial concerns often dominate NCM, reflecting fears about the affordability of raising children, including direct expenses like childcare and education, and indirect costs such as reduced career opportunities for parents. These concerns align with broader findings that economic insecurity leads individuals to prioritize stability over childbearing, delaying or forgoing parenthood altogether [41]. Italian society traditionally values strong familial support systems and emphasizes high-quality caregiving for children. This cultural expectation may exacerbate financial concerns, as parents feel compelled to meet high standards of care and education. Consequently, the anticipated financial and social pressures contribute to stronger NCM, particularly among individuals with precarious economic circumstances [42].

The “Physical Suffering and Body-Image Concerns” factor reflects worries about the physical discomforts of pregnancy and postnatal recovery, as well as changes in body image. In the original study, this factor explained 6% of the variance, marking it as a secondary but distinct dimension of negative motivations. In our study, this factor showed an increased contribution to variance explained and stronger internal consistency (Cronbach’s alpha = 0.77). This could be attributed to a heightened societal focus on aesthetics and health, particularly among women in urban areas. The recent literature corroborates these observations. A 2021 study by Huang et al. found that concerns about body image and physical health are especially pronounced in cultures with a high media influence, aspects that can amplify anxieties related to pregnancy and postpartum recovery [43].

### 4.3. T-Test Comparisons for Income and Cultural Levels

*T*-tests were performed to compare CMS scores across different income and cultural levels to investigate how socioeconomic and cultural factors influenced childbearing motivations. These analyses aimed to determine whether variations in economic and educational status impacted the relative importance of the nine CMS factors. For income levels (Table 7), the *t*-test revealed significant differences in the “Financial Problems and Economic Constraints” factor of the negative childbearing motivations (NCM) subscale, indicating that participants from lower-income groups expressed stronger concerns about the economic burden of childbearing. This highlights the critical role of financial security in reproductive decision-making. Moreover, our findings indicate that individuals with higher income levels tend to feel more personally fulfilled and experience fewer financial and physical concerns regarding negative childbearing motivations. These results are consistent with recent studies on the desire to have children in Italy and personal financial conditions [44,45,46]. Additionally, changing gender roles and attitudes towards motherhood and fatherhood play a vital role in shaping childbearing decisions [47]. Therefore, sociocultural variables could significantly influence people’s priorities and perceptions of negative motivations to become parents, leading to differences in the variance explained by each component of the psychometric scale. In this study, Cohen’s d values indicate that income level has a small but significant effect on these factors.

*T*-test for educational levels (Table 8) highlight that individuals with medium education levels tend to score higher on the Socioeconomic Aspects, Continuity, Childbearing Burden and Immaturity, and Physical Suffering and Body-Image Concerns factors than those with high education levels. These differences suggest that educational attainment influences certain motivations and concerns related to childbearing, although the effect size was modest. Individuals with medium education levels may have greater concerns about the financial implications of childbearing, as they are more likely to occupy precarious or mid-level positions in the labor market compared to those with higher education levels. Research shows that economic insecurity disproportionately affects individuals with less education, heightening their perception of the costs of parenthood and reinforcing negative childbearing motivations [48]. The significant scores on the Continuity dimension among individuals with medium education levels may reflect a stronger emphasis on traditional familial and societal expectations. In fact, cultural norms often position parenthood as a means of preserving family lineage and social identity, which can be more pronounced among those with mid-level education [49]. Concerns about readiness for parenthood, including fears of immaturity and the burden of parenting responsibilities, may reflect a lack of access to resources and support systems. These fears are consistent with studies showing that individuals with medium education levels often face greater stress related to balancing work and family life, contributing to stronger NCM [50]. Greater concerns about physical suffering and body image among those with medium education levels could stem from limited access to accurate health information and healthcare resources. This aligns with findings that individuals with higher education levels often have better health literacy and a more proactive approach to managing pregnancy-related challenges [51].

The findings of this study provide valuable insights into the cultural and socioeconomic nuances influencing childbearing motivations. The cross-cultural analysis highlighted both the universality and variability of the nine CMS factors, with intrinsic motivations such as Personal Fulfillment showing consistency across contexts, while extrinsic factors like Socioeconomic Aspects and Financial Problems revealed marked cultural and economic sensitivities. The *t*-test comparisons further demonstrated that income and cultural levels significantly shape both positive and negative motivations, with financial concerns playing a dominant role among lower-income participants and social recognition being emphasized by individuals with higher cultural capital.

These results suggest that while the fundamental dimensions of childbearing motivations remain robust across cultures, their expression is shaped by local economic and social dynamics. This underscores the need for culturally and contextually tailored policies and interventions to support informed and fulfilling reproductive choices. Understanding these multifaceted influences is crucial for policymakers in Italy to address declining fertility rates and develop effective interventions to support families in making informed childbearing choices. These findings suggest that socioeconomic interventions should be part of a broader strategy to address individuals’ motivations and concerns related to childbearing. Addressing NCM in Italy in fact requires targeted policy interventions aimed at alleviating financial stress, such as providing affordable childcare, parental leave, and financial incentives for families. Such measures could help mitigate the psychological and practical barriers to childbearing, particularly in economically vulnerable populations [52]. 

Although data are collected from the general population, the CMS could also be used effectively in clinical populations, such as infertile individuals and couples. Although the analysis of the sociodemographic data in this study identified individuals with infertility issues, the small number of interviews did not allow for statistically significant results. In this sense, the CMS could also be useful for health professionals to make informed decisions about family planning in complex situations (e.g., female cancer or issues related to medically assisted reproduction in general).

Moving forward, these findings form a foundation for targeted applications of the CMS in diverse sociocultural settings, bridging the discussion to the broader implications and actionable recommendations outlined in the conclusion.

## 5. Conclusions

Our comprehensive findings validate the Italian version of the Childbearing Motivations Scale (CMS) as a robust and culturally adapted tool for measuring childbearing motivations. This tool maintains consistency with the original English version in terms of psychometric properties, measurement characteristics, and factor structure. The Italian CMS reliably captures both positive and negative dimensions of childbearing motivations, encompassing crucial factors that align with contemporary theoretical and empirical insights on fertility behaviors.

The validation study incorporates more recent theoretical and empirical insights on fertility motivations, particularly those related to economic instability, delayed parenthood, and evolving gender roles in Italy. This study in fact integrates updated perspectives on childbearing motivations, shedding light on the unique challenges posed by Italy’s socioeconomic environment.

Financial concerns emerged as a critical driver of negative childbearing motivations in the Italian context, resonating with prior research that identified economic factors as a significant barrier to parenthood. This finding resonates with existing research linking economic uncertainty to delayed parenthood and declining fertility rates in Italy [53].

Furthermore, the CMS offers valuable insights into the psychological and emotional dimensions of childbearing, particularly in the interplay between intrinsic and extrinsic motivations. These findings underscore the multidimensional nature of childbearing motivations and expand our understanding of the reproductive motivations, encompassing not only economic and social influences but also emotional and identity-driven factors [54].

Socioeconomic status (SES) and educational attainment were identified as significant moderators of childbearing motivations. Higher SES participants exhibited stronger positive motivations, particularly those related to personal fulfillment and relational dynamics, while reporting fewer negative concerns such as financial constraints or physical suffering. Conversely, individuals with medium levels of educational attainment perceived greater socioeconomic benefits of parenting but expressed heightened worries about parental immaturity and physical challenges, suggesting the nuanced influence of education on reproductive attitudes. In addition to its psychometric validation, the CMS offers significant practical utility [55].

### 5.1. Applications and Policy Implications

The validated Italian CMS has profound implications for research, clinical practice, and policymaking. In the Italian context, where fertility rates remain persistently low, a detailed understanding of childbearing motivations can aid in crafting targeted interventions. For instance, reproductive counseling could benefit from incorporating CMS assessments to guide individuals and couples through family planning decisions that align with their values and circumstances. In clinical settings, the scale holds potential for addressing fertility-related challenges, such as infertility and medically assisted reproduction, by enabling a personalized exploration of patients’ motivations and concerns. Moreover, the CMS could serve as a predictive tool for analyzing reproductive intentions and behaviors within diverse sociocultural contexts. In fact, it could help identify individuals at risk of unrealistic expectations or psychosocial difficulties during the transition to parenthood, facilitating early interventions to foster emotional and relational adjustment [56,57,58,59,60,61]. Policymakers can leverage these insights to address systemic barriers to parenthood, such as financial instability, and to promote equitable access to support services for prospective parents [62].

### 5.2. Recommendations for Future Research

To further enhance the generalizability and applicability of the CMS, future studies should prioritize the following:Diverse Populations: expand research to include clinical samples, such as individuals undergoing fertility treatments or coping with infertility, and cross-cultural comparisons to evaluate the CMS’s adaptability across different societal contexts.Longitudinal Approaches: conduct longitudinal studies to explore how childbearing motivations evolve over time in response to shifting economic, social, and relational dynamics.Enhanced Sampling Strategies: recruit more representative samples, particularly from underrepresented demographics, including rural populations and non-student groups.Integrated Frameworks: investigate the interaction between motivations for childbearing and other psychological constructs, such as attachment styles and coping mechanisms, to develop a more comprehensive understanding of reproductive decision-making.

In conclusion, the Italian CMS represents a significant advancement in understanding the complex interplay of motivations influencing childbearing decisions. Its multidimensional framework captures a broad spectrum of positive and negative influences, offering researchers, clinicians, and policymakers a nuanced perspective on fertility trends. By identifying both facilitators and barriers to parenthood, the scale can guide efforts to address Italy’s demographic challenges, supporting individuals and families in making informed and fulfilling reproductive choices.

## Figures and Tables

**Table 1 ijerph-22-00186-t001:** Sociodemographic Characteristics of the Sample.

Characteristic	Frequency (*n*)	%
Sex		
Male	141	27%
Female	379	72.6%
Other N/R	2	0.4%
Nationality		
Italian	509	97.5%
Foreign	9	1.7%
Other N/R	4	0.8%
Education		
High school or professional degree	322	61.7%
Bachelor’s degree or higher	194	37.2%
Only a high school diploma	3	0.6%
Other N/R	3	0.6%
Sexual Orientation		
Heterosexual	465	89.1%
Bisexual	42	8%
Homosexual	11	2.1%
Other N/R	4	0.8%
Employment		
Student	339	64.9%
Employed with permanent contract	80	15.3%
Employed with fixed term contract	37	7.1%
Self Employed	39	7.5%
Unenployed/Looking for first job	9	1.7%
Homely/Housewife	3	0.6%
Retired	1	0.2%
Other N/R	14	2.7%
Socio-Economic Status (SES)		
Low	330	63.3%
Medium	157	30.1%
High	35	6.6%
Marital Status		
Single	399	76.4%
Cohabitating	73	14%
Married	42	8%
Divorced	6	1.1%
Separated	1	0.2%
Widoved	1	0.2%
Living Arrangement		
With family or friends	345	66.1%
With Partner	118	22.6%
Alone	39	7.5%
Other	20	3.8%

**Table 2 ijerph-22-00186-t002:** Positive childbearing motivations subscale: items’ descriptive statistics, internal consistency, and factor loadings. Note: Adapted with permission from [20].

Item	M (SD)	%	F1	F2	F3	F4	α
Socioeconomic Aspects		46.79					0.91
Economic Support	2.09 (1.40)		0.68				
Responsability Affirmation	2.31 (1.33)		0.88				
Adult Affirmation	2.46 (1.30)		0.66				
Social Valorization	2.00 (1.15)		0.91				
Moral Obligation	1.94 (1.22)		0.83				
Family Expectations	2.15 (1.18)		0.86				
Gender Role	2.17 (1.27)		0.46				
Couple Recognition as a Family	2.15 (1.21)		0.80				
Personal Fulfillment		10.79					0.87
Biological Clock	2.51 (1.21)			0.60			
Pregnancy Experience	2.60 (1.35)			0.81			
Maternal or Paternal Instinct	3.20 (1.32)			0.92			
Creating a Personality	3.47 (1.21)			0.88			
Creating a Family	3.62 (1.21)			0.92			
Blood Ties	2.41 (1.28)			0.44			
Life Meaning	2.86 (1.83)			0.46			
Feeling Important for a Child	3.15 (1.29)			0.46			
Continuity		4.41					0.85
Familial Lineage	2.18 (1.19)				0.77		
Family’s Name	2.20 (2.25)				0.94		
Family’s Relationships	2.71 (1.23)				0.49		
Family’s Heritage	2.30 (1.20)				0.63		
Family’s Value	3.00 (1.28)				0.51		
Family Spirit	3.35 (1.22)				0.53		
The Couple’s Relationship		3.94					0.80
Strengthening Partnership Ties	2.82 (1.31)					0.71	
Fulfilling Partner’s Project	2.17 (1.19)					0.75	
Growing as a Couple	2.31 (1.26)					0.45	
Fulfilling a Shared Project	3.22 (1.28)					0.48	

**Table 3 ijerph-22-00186-t003:** Negative childbearing motivations subscale: items’ descriptive statistics, internal consistency, and factor loadings. Note: Adapted with permission from [20].

Item	M (SD)	%	F1	F2	F3	F4	F5	α
Childbearing Burden Immaturity		3.01						0.70
Constant Worry	3.05 (1.05)		0.42					
Lifelong Responsability	3.28 (1.25)		0.41					
Constant Needs of a Child	3.21 (1.18)		0.67					
Childcare Labor	3.36 (1.22)		0.31					
Concerns Parental Qualities	2.58 (1.12)		0.64					
Concern Parental Preparedness	3.01 (0.60)		0.60					
Social and Ecological Worry		3.92						0.77
Worry about the future	3.86 (1.01)			0.64				
Enviromental Degradation	3.33 (1.11)			0.70				
Social Danger	3.35 (1.08)			0.90				
Deviant Trajectories	3.16 (1.06)			0.39				
Marital Stress		5.67						0.83
Constraints for Couple’s Proximity	2.34 (1.05)				0.91			
Constraints for Couple’s Autonomy	2.63 (1.07)				0.73			
Constraints for Couple’s Routines	2.76 (1.04)				0.51			
Fear of Couple’s Separation	2.02 (1.01)				0.62			
Financial Problems Economic Constraints		38.13						0.87
Increased Expenses	3.22 (1.06)					0.85		
Financial Sacrifices	3.03 (1.09)					0.78		
Financial Difficulties	3.04 (1.08)					0.67		
Constraints for Financial Well Being	2.55 (1.08)					0.56		
Physical Suffering Body Image Concerns		2.64						0.76
Family’s Value	2.56 (1.21)						0.72	
Family Spirit	2.60 (1.15)						0.51	

**Table 4 ijerph-22-00186-t004:** Indices for Confirmatory Factor Analysis (CFA).

Model	χ2	df	P	χ2/df	CFI	RMSEA [90% CI]
Positive Childbearing Motivations						
Four-Factor Model	788.32	255	<.001	3.1	0.98	0.08 [0.08, 0.09]
Negative Childbearing Motivations						
Five-Factor Model	479.11	168	<.001	2.85	0.96	0.08 [0.07, 0.09]

CFI comparative fit index, RMSEA root-mean-square error of approximation 90% CI = 90% confidence interval. All χ2 tests were significant at *p* < 0.001.

**Table 5 ijerph-22-00186-t005:** Intercorrelations between the factors of the positive childbearing motivations subscale and the negative childbearing motivations subscale. Note: Adapted with permission from [20].

Variable	1	2	3	4	5	6	7	8	9
1. Socioeconomic Aspects									
2. Personal Fulfillment	0.67 ***								
3. Continuity	0.81 ***	0.78 ***							
4. The Couple’s Relationship	0.71 ***	0.70 ***	0.70 ***						
5. Childbearing Burden	0.30 ***	0.18 ***	0.25 ***	0.25 ***					
6. Social and Ecological Worry	0.16 ***	0.14 ***	0.16 ***	0.16 ***	0.69 ***				
7. Marital Stress	0.27 ***	0.08 ***	0.19 ***	0.26 ***	0.52 ***	0.46 ***			
8. Financial Problems	0.16 ***	−0.02	0.10 ***	0.12 ***	0.55 ***	0.59 ***	0.60 ***		
9. Physical Suffering	0.27 ***	0.12 ***	0.24 ***	0.25 ***	0.59 ***	0.51 ***	0.54 ***	0.50 ***	

Note: *** *p* < 0.001.

**Table 6 ijerph-22-00186-t006:** Correlations between age, relationship’s duration, sex, and CMS dimensions.

	F1	F2	F3	F4	F5	F6	F7	F8	F9
Age	−0.08	0.07	−0.03	−0.05	−0.21 **	−0.12 **	−0.10 **	−0.18 **	−0.26 **
Relationship Duration	−0.04	0.10	0.1	−0.04	−0.20 **	−0.10	−0.12 *	−0.20 **	−0.16 **
Sex	−0.05	0.06	0.08 *	0.08	−0.21 **	−0.23 **	−0.10 *	−0.10 *	−0.22 **

Note: * *p* < 0.05; ** *p* < 0.01.

**Table 7 ijerph-22-00186-t007:** *T*-test comparison of the CMS dimensions as a function of income level [Low (1)—Medium/High (2)].

	Income Level	N	Mean	SD	*t*-Value	*p*-Value	Cohen’s D
Socioeconomic Aspects	1	323	17.56	8.57	1.19	0.23	0.11
	2	168	16.63	7.52			
Personal Fulfillment	1	325	23.35	7.58	−2.02	0.04 *	−0.19
	2	168	23.78	7.24			
Continuity	1	326	15.82	6.06	0.24	0.80	0.02
	2	171	15.69	5.24			
The Couple’s Relationship	1	324	10.56	4.18	0.15	0.88	0.14
	2	168	10.50	3.87			
Childbearing Burden	1	316	18.72	4.63	1.67	0.09	0.16
	2	168	18.01	4.14			
Social and Ecological Worry	1	323	13.80	3.44	0.95	0.34	0.09
	2	169	13.50	2.81			
Marital Stress	1	317	9.82	3.39	0.39	0.69	0.03
	2	169	9.70	3.55			
Financial Problems	1	322	12.17	3.68	2.67	0.008 **	0.25
	2	169	11.24	3.57			
Physical Suffering	1	322	8.32	2.92	2.99	0.003 **	0.28
	2	169	7.51	2.89			

Note: * *p* < 0.05; ** *p* < 0.01.

**Table 8 ijerph-22-00186-t008:** *T*-test comparison of the CMS dimensions as a function of education level [Medium (1)—High (2)].

	Education Level	N	Mean	SD	*t*-Value	*p*-Value	Cohen’s D
Socioeconomic Aspects	1	314	17.76	8.64	2.19	0.03 **	0.20
	2	192	16.13	7.24			
Personal Fulfillment	1	318	23.64	7.68	−0.74	0.46	−0.06
	2	190	24.15	7.26			
Continuity	1	319	16.02	5.98	1.55	0.12	0.14
	2	192	15.20	5.38			
The Couple’s Relationship	1	316	10.60	4.28	0.57	0.56	0.05
	2	192	10.39	3.73			
Childbearing Burden	1	308	18.82	4.41	2.18	0.03 *	0.20
	2	191	17.93	4.46			
Social and Ecological Worry	1	313	13.82	3.22	1.04	0.29	0.09
	2	193	13.51	3.28			
Marital Stress	1	311	9.75	3.40	−0.001	0.99	0.00
	2	190	9.75	3.29			
Financial Problems	1	312	11.95	3.65	1.06	0.28	0.09
	2	193	11.06	3.66			
Physical Suffering	1	314	8.35	2.90	2.55	0.01 **	
	2	192	7.67	2.86			

Note: * *p* < 0.05; ** *p* < 0.01.

## Data Availability

The data used in this study are from the Dipartimento di Scienze della Salute (DSS), Università degli Studi di Firenze, Via San Salvi 12, 50135 Firenze. The data generated and/or analyzed as part of this study are available upon request from the corresponding author.

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
