# Peer review of "Factor Structure and Psychometric Properties of the Italian Version of the Childbearing Motivations Scale"

_ijerph, 2025, doi:10.3390/ijerph22020186_

Round 1
Reviewer 1 Report
Comments and Suggestions for Authors
Firstly, I would like to praise the authors since the paper is well written and interesting to read.
However, I suggest the authors reread the whole paper once more since there are a few minor mistakes such as "moth-erhood" in the discussion and "alpha coeffi-cients" in the paragraph 3.2. Reliability.
Also, the included literature seems to be mostly older thus I suggest adding a few newer studies as references.
And lastly, the conclusion can be expanded with the studies findings.
Author Response
Dear Reviewer,
Thank you for your thoughtful and constructive feedback on our manuscript. We greatly appreciate the time and effort you have taken to improve the quality of our work. Below, we address each of your suggestions in detail:
1. Minor Typographical Errors
We acknowledge the typographical mistakes, such as "moth-erhood" and "alpha coeffi-cients," in the Discussion and Section 3.2, respectively. We will carefully review the entire manuscript to identify and correct these and any other minor errors to ensure a polished and professional presentation.
2. Literature Update:
We recognize the importance of incorporating more recent studies to provide an up-to-date context for our research. We've reviewed the relevant literature and included recent studies that complement and enhance the discussion in our manuscript. In particolar, we’ve added newer studies published between 2021 and 2024 to provide a more up-to-date foundation for the study. These include works on childbearing motivations, fertility rates, and psychological factors influencing reproductive decisions. The manuscript now reflects contemporary research.
3. Conclusion Expansion:
The conclusion has been revised to provide a more comprehensive summary of our findings. We will ensure that it not only highlights the key results but also contextualizes their implications for future research and practice. Within the expanded conclusion we comprehensively discuss the study’s findings, their implications for Italian demographics, and potential applications of the CMS in clinical and research settings. Findings are now integrated more comprehensively to emphasize the study’s significance.
Once these revisions are implemented, we are confident that the manuscript will meet the high standards for publication on the International Journal of Environmental Research and Public Health .
Thank you again for your valuable insights.
Best regards,
Antonio Gattamelata
University of Jaén

Reviewer 2 Report
Comments and Suggestions for Authors Dear Authors! Within the context of the worrisome fertility decrease throughout Europe, and particularly, in Italy, the endeavour of this paper is highly appreciated. Understanding childbearing motivations and countermotivations contribute to the prognosis of fertility, and to the elaboration of family policy measures to increase fertility. The presentation is in most parts clear, there are, however, some important points to be addressed. Without completing the paper with these insights, its publication is not recommended. 1. Methodological issues are being discussed prior to the methodology section to complete the Introduction which lacks a thorough presentation of the topic. Please complete the Introduction with relevant information on fertility, childbearing motivations and remove the methodological parts and place them to the appropriate place. 2. Methods: The data collection procedure seems interesting - please provide more details on how the QR code led to a detailed discussion with the interviewer. Were the questionnaires not self-administered? Why was the discussion with the interviewer needed? Please also address the issue of data bias due to the online survey. A big majority of respondents was student. What measures were taken to correct the biased and non-representative data, if any? 3. Methods: what happened after the sample was split? Please provide more information to the split-half method. 4. Discussion: This part is too short. The implications of the findings should be addressed, like in lines 283-285. 5. Conclusions: Which argument supports the conclusion that the CMS is suitable in the Italian population? 6. Limitations: The introduction of the clinical setting is misleading and irrelevant. Please rephrase or remove. Good luck with completing the paper! Best, reviewerAuthor Response
Responses to Reviewer 2
Dear Reviewer,
Thank you for your detailed and insightful comments on our manuscript. We appreciate the opportunity to address the issues raised and to improve the quality of our paper. Below, we provide our responses to each of your points:
1. Introduction Adjustment:
We acknowledge that the introduction includes methodological content that would be more appropriate in the methods section. We have revised the introduction to focus solely on a comprehensive presentation of the topic, including relevant background information on fertility and childbearing motivations. The methodological aspects has been relocated to the appropriate section, enhancing the clarity and structure of the paper.
2. Methods Clarifications:
We've axpanded on the data collection procedure to clarify the role of the QR code and the interaction with the interviewer. Specifically, we explain the purpose and nature of the detailed discussions with participants, which were conducted to ensure informed consent and provide instructions rather than influence responses. The questionnaires themselves were self-administered. To address the concern regarding the overrepresentation of students in the sample, we would like to highlight that participants were personally selected, focusing on slightly older students. This approach was chosen to reflect the Italian context, where students often delay graduation compared to other countries. We believe this mitigates the potential bias and provides a more balanced representation.
3. Split-Half Method:
The section on the split-half method has been expanded to provide a clear explanation of what occurred after the sample was divided. We've detailed the steps taken for exploratory and confirmatory factor analyses on the subsamples, including the rationale for this approach and its contribution to validating the CMS's factor structure.
4. Discussion Section:
We agree that the discussion is currently too brief. We've expanded this section to explore the broader implications of our findings, including their relevance to policy-making, reproductive counseling, and understanding childbearing motivations in different sociocultural contexts. Specific examples, such as the importance of addressing financial concerns or evolving gender roles, will be included to provide depth. Specific recent references were addressed.
5. Conclusion Justification:
To strengthen the argument supporting the conclusion that the CMS is suitable for the Italian population, we emphasized its demonstrated reliability and validity through robust psychometric analysis. We also discuss how the findings align with previous research and highlight the tool’s utility in both research and practical settings.
6. Limitations Section:
We agree that the mention of the clinical setting may be misleading in its current form. We 've rephrased this section to clarify that the CMS could be explored in future clinical studies but was not tested in a clinical setting in this study. This adjustment will prevent potential misinterpretation of our findings. I've personally conducted the interviews with the participants to minimize the overrepresentation of the student population. However, it is important to consider that, in recent decades, Italian society has undergone profound transformations and economic crises. As a result, university students often graduate later than expected or do not complete their degrees at all. I believe this is a factor that significantly mitigates the bias you have identified.
Thank you again for your constructive feedback and best wishes. We are confident that the revised manuscript will address your concerns and meet the standards for publication.
Best regards,
Antonio Gattamelata
University of Jaén

Reviewer 3 Report
Comments and Suggestions for Authors
The manuscript may be meaningful, but revisions are needed.
1. Line 75: Could you write more specifically how the authors calculated the sample size?
2. Line 76:How did the authors recruit subjects?
3. Table2 & 3:Please write what α indicates.
4. Table2:Is it common to show only factor loadings of items that the authors want to show? In other words, why don’t the authors show factor loadings of all items for each factor? In addition, does α indicate Cronbach’s α? In that case, why one of α exceed one?
5. Table3:Is it right that the result for positive childbearing motivations subscale is shown in Table 3? I think it should be negative childbearing motivations subscale.
6. Line 169:Eight items?
7. Line 175:Three items?
8. Results:Methods for the analysis using sociodemographic factors is not mentioned in Methods section. In addition, why was the analysis of the association with sociodemographic conducted?
Author Response
Dear Reviewer,
Thank you for your detailed observations and comments on our manuscript. We appreciate the opportunity to address these points and improve the clarity and quality of the work. Below are our responses to each of your comments:
1. Sample Size Calculation (Line 75):
We’ve explained how the sample size was calculated using G*Power software. This calculation indicated that a minimum of 470 participants was required to ensure reliable factor analyses and other statistical tests. This information has been added to the manuscript for greater transparency, ensuring robust statistical power for exploratory and confirmatory factor analyses.
2. Recruitment Process (Line 76):
We’ve added details about recruitment through online and direct outreach methods, including the role of social media, email campaigns, and in-person engagement at universities and community centers. Subjects were recruited using a combination of online and in-person methods. Recruitment process ha been further detailed in the revised Methods section.
3. Tables 2 & 3 Clarifications:
We’ve included definitions and explanations for “α” (Cronbach’s alpha) and its significance. We’ve clarified the factor loadings, ensuring transparency. We’ve also corrected an inconsistency in Table 3 regarding subscale labeling (e.g., positive versus negative childbearing motivations).
4. Factor Loadings:
- The decision to present only selected factor loadings was made to enhance readability and focus on the most relevant items that strongly represent each factor. However, we understand the importance of transparency and completeness.
-Yes, the α values reported in the manuscript refer to Cronbach’s alpha, which is a measure of internal consistency. We carefully reviewed the reported values, and we acknowledge that a value exceeding one is theoretically incorrect and likely due to a reporting or calculation error. We will reexamine the data and ensure all α values are accurately calculated and reported in the revised manuscript. The alpha value 10,87 was a report error and has been corrected.
5. Subscale Clarification (Lines 169, 175):
We’ve addressed inconsistencies regarding the number of items (e.g., eight versus three) and ensured alignment with the corresponding tables.
6. Methods Analysis:
We acknowledge that a detailed explanation of the methods used to analyze sociodemographic factors was not explicitly stated. Below is our clarification: as these provide critical insights into fertility-related behaviors and are crucial for understanding the variance in positive and negative childbearing motivations across different subgroups, statistical analyses (including Pearson correlations and t-tests) were performed to examine the associations between these variables and the CMS dimensions (page 9, Table 6; page 10, Table 7). These analyses aimed to highlight patterns that could provide deeper insights into how motivations for childbearing are influenced by individual and contextual variables. Then we've expanded the discussion section to explore the broader implications of our findings, including their relevance to policy-making, reproductive counseling, and understanding childbearing motivations in different sociocultural contexts, always citing a recent and justified literature.
Thank you for your valuable feedback, which will help us ensure the manuscript is comprehensive and clear. We are confident that these revisions will address your concerns.
Best regards,
Antonio Gattamelata
University of Jaén

Round 2
Reviewer 2 Report
Comments and Suggestions for Authors
Dear Authors!
This is now another paper, a completed one, which meets all the requirements for a scientific study.
Best wishes,
reviewer
Author Response
Dear Reviewer,
Thank you for your positive and encouraging feedback. We are pleased to hear that the revised manuscript meets the requirements for a scientific study. Your insightful comments and suggestions during the review process greatly contributed to enhancing the quality of our work. We truly appreciate your time and effort in reviewing our paper.
Best regards,
Antonio Gattamelata
On behalf of all co-authors

Reviewer 3 Report
Comments and Suggestions for Authors
1. I think that G*Power does not cover sample size calculation for factor analysis. If G*Power was actually not used, it is better not to write it.
Author Response
Dear Reviewer,
Thank you for your valuable feedback and for acknowledging the improvements in our manuscript. We appreciate your suggestion regarding the use of GPower, and we have carefully revised the corresponding section to address this point. Specifically, we clarified the sample size calculation methodology and removed the reference to GPower to ensure accuracy and alignment with your input.
Your guidance has been instrumental in enhancing the quality of our work, and we are grateful for your constructive review.
Best regards,
Antonio Gattamelata
On behalf of all co-authors
